# Biodegradability Assessment of Prickly Pear Waste–Polymer Fibers under Soil Composting

**DOI:** 10.3390/polym15204164

**Published:** 2023-10-20

**Authors:** Zormy Nacary Correa-Pacheco, Silvia Bautista-Baños, José Jesús Benítez-Jiménez, Pedro Ortega-Gudiño, Erick Omar Cisneros-López, Mónica Hernández-López

**Affiliations:** 1Centro de Desarrollo de Productos Bióticos, Instituto Politécnico Nacional, Carretera Yautepec-Jojutla, Km. 6, Calle CEPROBI, No. 8, San Isidro, Yautepec 62731, Morelos, Mexico; sbautis@ipn.mx (S.B.-B.); mohernandezl@ipn.mx (M.H.-L.); 2Instituto de Ciencia de Materiales de Sevilla, CSIC-Universidad de Sevilla, Avda. Américo Vespucio 49, Isla de la Cartuja, 41092 Sevilla, Spain; benitez@icmse.csic.es; 3Departamento de Ingeniería Química, Centro Universitario de Ciencias Exactas e Ingenierías, Universidad de Guadalajara, Blvd. Gral. Marcelino García Barragán #1451, Guadalajara 44430, Jalisco, Mexico; erick.cisneros@academicos.udg.mx

**Keywords:** degradation time, morphology, thermal properties, mechanical properties, weight loss

## Abstract

Nowadays, solving the problems associated with environmental pollution is of special interest. Therefore, in this work, the morphology and thermal and mechanical properties of extruded fibers based on polylactic acid (PLA) and poly(butylene adipate-co-terephthalate) (PBAT) added to prickly pear flour (PPF) under composting for 3 and 6 months were evaluated. The highest weight loss percentage (92 ± 7%) was obtained after 6-month degradation of the PLA/PBAT/PPF/CO/AA blend, in which PPF, canola oil (CO), and adipic acid (AA) were added. Optical and scanning electron microscopy (SEM) revealed structural changes in the fibers as composting time increased. The main changes in the absorption bands observed by Fourier transform infrared spectroscopy (FTIR) were related to the decrease in -C=O (1740 cm^−1^) and -C-O (1100 cm^−1^) groups and at 1269 cm^−1^, associated with hemicellulose in the blends with PPF. Differential scanning calorimetry (DSC) showed an increase in the cold crystallization and melting point with degradation time, being more evident in the fibers with PPF, as well as a decrease in the mechanical properties, especially Young’s modulus. The obtained results suggest that PPF residues could promote the biodegradability of PLA/PBAT-based fiber composites.

## 1. Introduction

Currently, compatibilized polymer blends based on biodegradable polymers added to recycled agriculture crop residues have gained special attention in polymer applications [1,2]. In packaging applications, PBAT poly (butylene adipate-co-terephthalate) is considered a good candidate to improve the toughness of PLA (polylactic acid) because the later has a low heat-resistance capacity, which limits its usefulness in food packaging. Therefore, PLA and PBAT complement each other. However, the PLA/PBAT blend is considered immiscible; therefore, having good interface compatibility between PLA and PBAT is essential to obtaining a blend with satisfactory properties. Both are biodegradable polymers that have recently become popular due to their potential application in the packaging field [3,4].

*Opuntia ficus-indica*, commonly known as prickly pear, belongs to the Cactaceae family and consists of four main parts: cladodes (succulent stems), flowers, fruit, and seeds. This is a plant that easily adapts to the environment. The cladodes contain mucilage and cellulosic fibers (80–95%), carbohydrates (3–7%), fibers (1–2%), and small amounts of protein (0. 5–1%) [5,6].

Scaffaro et al. [7] obtained a lignocellulosic flour by grinding *O. ficus-indica* cladodes to prepare extruded biocomposites with PLA. They found a notable increase in stiffness, up to 135% of the polymer matrix, and, at the same time, a decrease in tensile strength, elongation at break, and toughness. Regarding the size of *O. ficus-indica*, it was found that the larger the size of the filling, the greater the stiffness and fragility due to the contact area between the filler and the polymer matrix. On the other hand, Spiridon et al. [8] studied the effect of the incorporation of lignin microparticles into PLA by extrusion. From the morphology, a homogeneous distribution of lignin into the PLA matrix without holes, cracks, and aggregations was observed by scanning electron microscopy (SEM). Young’s modulus increased and the tensile strength decreased with the addition of 7% lignin compared to the PLA polymer matrix. When 15% lignin was incorporated into the PLA matrix, there was an increase in the tensile strength; therefore, the lignin content was an important factor in controlling the mechanical properties of the composites.

The use of compatibilizers such as adipic acid (AA) incorporated into blown-film polymer blends has been evaluated by Kim et al. [9]. They found that AA improves the flexibility of PLA and its compatibility with PBAT, resulting in enhanced mechanical properties. Differential scanning calorimetry (DSC) revealed that AA exerted a plasticizing effect, lowering the transition peaks with the presence of one melting peak. SEM revealed phase-separated structures and voids. However, when the AA content increased, a continuous phase morphology was seen due to the improvement in the interfacial adhesion. The mechanical properties showed a decrease in the tensile strength and an increase in the elongation at break with the addition of AA.

Studying the biodegradability of polymer blends combined with additives and agricultural waste is necessary due to their application as compostable materials. Borelbach et al. [10] studied the degradation of biodegradable polymer fibers from PLA under a soil environment at 23 °C for 12 weeks. After this time, no degradation was observed. After 2 weeks, the tensile strength decreased. However, after 8 weeks it increased. Fourier transform infrared spectroscopy (FTIR) and DSC showed no changes over the degradation time. Ramle et al. [11] employed cellulose from bamboo as a reinforced filler in PLA/PBAT blends, studying their degradation in soil. The cellulose content in the prepared films was 0%, 3%, 6%, and 9%. The weight loss was evaluated, and the results showed a value of 12.39% for the 9% PLA/cellulose film, followed by the PLA/PBAT/cellulose blend at the same concentration (9.69%). Moreover, a higher cellulose content enhanced the biodegradability after a period of 15 days. Rebelo et al. [12] simulated an outdoor accelerated weathering environment for a commercial PLA/PBAT blend (BioFlex^®^) with different organic acids (acetic and its derivatives). They reported that the organic acids accelerated the blend´s degradation. The results of FTIR before and after degradation, after 200 h by UV, did not show significant differences. Only small changes in the band at 1741 cm^−1^ corresponding to C-O stretching were observed. SEM revealed cracks and holes after degradation. DSC before and after degradation showed similar events. However, in the presence of the acetic and its derivatives, some chain scission occurred in the polymer blend with small crystalline peaks due to oligomer formation. In terms of mechanical performance, the tensile strength and elongation at break decreased in the PLA/PBAT blend without additives.

The aim of this work was to study the biodegradation process in soil of PLA/PBAT polymer fibers with the addition of prickly pear flour as agro-industrial waste, adipic acid as a compatibilizer, and canola oil as a plasticizer.

## 2. Materials and Methods

### 2.1. Materials

To make the fibers, the PLA, Biopolymer 7001D (Ingeo™, NatureWorks^®^ LLC, Minnesota, MN, USA), with a specific gravity of 1.24 g cm^−3^ and a melt flow index of 6 g/10 min (210 °C, 2.16 kg), and the PBAT, Ecoflex^®^ F Blend C1200 (BASF SE, Ludwigshafen, Germany), with a specific gravity of 1.25–1.27 g cm^−3^ and a melt flow index of 2.7–4.9 g/10 min (190 °C, 2.16 kg), were used. The neat polymer pellets were dried at 60 °C for 24 h in a conventional oven before use. A prickly pear flour (PPF) from waste cladodes collected in the municipality of Tlalnepantla, Morelos, Mexico, was incorporated into the extruder by an emulsion that contained canola oil (CO) (Valley Foods^®^, Monterrey, Mexico) and Tween 80 (Hycel^®^, Zapopan, Jal., Zapopan, Mexico). Adipic acid (AA) (Meyer^®^, Mexico City, Mexico) was used as a compatibilizer.

### 2.2. Methods

#### 2.2.1. PPF Preparation

To obtain the PPF, the methodology of Scaffaro et al. [7] was followed with some modifications. First, the cladodes were rinsed with potable water to reduce surface contamination and cut into 2 × 2 cm pieces. Then, they were oven-dried (Binder^®^ FD23ULE2, Tuttlingen, Germany) at 60 °C for 24 h. After that, the sample was ground with a blender (Osterizer^®^ 465-43, Mexico city, Mexico) for 1 min. Then, the flour was passed through a 50-mesh sieve (100 µm) and sealed and stored at room temperature.

#### 2.2.2. Fiber Preparation

Fibers were prepared using a twin-screw extruder (Process 11, Thermo Scientific™, Waltham, MA, USA). The temperature profile was 160/170/170/180/180/190/190/160 °C from the feeding zone to the die. The outlet nozzle was 2.5 mm. The fibers were cooled in water. The composition of the polymer blend was PLA/PBAT 60/40 based on previous studies [13]. To add the compatibilizer and the plasticizers to the polymer blend in the extruder, an emulsion based on Tween 80 (0.5%), PPF (3%), CO (4%), and AA (7%) was prepared considering two temperatures, 27 ± 2 °C and 60 ± 2 °C, which was added to the extruder using a peristaltic pump (MasterFlex C/L, Cole-Parmer, Vernon Hills, IL, USA). The feeding speed was 0.15 mL/min. In total, six fibers were extruded: neat PLA and PBAT, PLA/PBAT, PLA/PBAT/CO, PLA/PBAT/PPF/CO/AA without emulsion heating or temperature (PLA/PBAT/PPF/CO/AA) and without emulsion heating (PLA/PBAT/PPF/CO/AA_T). The fiber composition is explained in Table 1.

#### 2.2.3. Biodegradability Tests

The biodegradability of the fibers knitted as nets in soil was assessed using the methodology of López et al. [14] with some modifications. A blend of litter/soil/donkey manure in a proportion of 6/2/1 (*v*/*v*) was allowed to settle for 30 days with constant watering. Then, the nets were buried at a depth of 5 cm from the surface. The composting time was 3 and 6 months. After that, the nets were removed from the soil, cleaned, and weighed using a portable digital scale (OHAUS, Parsippany, NJ, USA) to determine the percentage of weight loss. This variable was calculated between the initial and the final net weight after composting.

#### 2.2.4. Optical and SEM Observations

The morphology of the fibers from the nets was observed throughout the composting process using a zoom lens (Edmund Optics, VZM 200i, York, UK) and an optical microscope (Nikon, Optiphot2, Tokyo, Japan), both coupled to a digital camera. Higher-resolution imaging was performed using a SEM (Hitachi, S-4800, Tokyo, Japan) operated at 15 kV. In this latter case, the fibers were previously coated with a thin layer of gold.

#### 2.2.5. Fourier Transform Infrared Spectroscopy–Attenuated Total Reflection (FTIR-ATR)

The ATR spectra of the neat fiber polymers and blends before and after composting were recorded using an FT-IR spectrometer (Thermo Nicolet, IS50, Minneapolis, MN, USA) equipped with an attenuated total reflection accessory (Thermo Nicolet, Smart Performer, USA) with a ZnSe single reflection crystal. A resolution of 4 cm^−1^ was used and 32 scans within the range of 4000–650 cm^−1^ were accumulated.

#### 2.2.6. Differential Scanning Calorimetry (DSC)

The DSC measurements were carried out using a DSC-Q20 (TA Instruments, Wakefield, MA, USA) before and after composting. The instrument was calibrated with an indium standard and non-hermetic aluminum crucibles were used.

#### 2.2.7. Mechanical Properties

The uniaxial tensile tests of the fibers were conducted with a universal machine (MTS, Criterion C42, Huntsville, AL, USA) equipped with a 50 N load cell (MTS, LSB.501, USA) using an initial crosshead distance of 10 mm and a deformation rate of 5 mm/min.

#### 2.2.8. Statistical Analysis

To establish the differences between the formulations, a one-way analysis of variance with Tukey’s mean comparison test (*p* ≤ 0.05) was performed. InfoStat software, version 2020 (National University of Córdoba, Córdoba, Argentina) was used.

## 3. Results and Discussion

### 3.1. Biodegradability Tests

According to the results, the highest weight loss was obtained by the PLA/PBAT/PPF/CO/AA with and without temperature after 6 months. For neat polymers, after 3 months, PLA showed a higher weight loss compared to PBAT. This could be attributed to the ease of degradation of the aliphatic backbone of PLA compared to the aromatic component in the backbone of PBAT [15]. This behavior was reported by Carbonell-Verdu et al. [16] for PLA in PLA/PBAT films. However, the final value (6 months) was almost the same for both polymers. PLA and PBAT degradation efficiency is related to the difference in polymer structure and the action of microorganisms in the soil [17]. Some authors have demonstrated for PLA/PBAT blends that, during composting, there is a selectivity to degrade the PLA phase with the formation of a porous 3D network [18] as will be discussed in the Section 3.2. In the literature, it has been reported that the lower weight loss of PBAT can be attributed to the presence of benzene rings [19]. During degradation, microorganisms attack the ester bond in PLA and PBAT, with the generation of small molecules or oligomers, which are subsequently metabolized by these microorganisms [17]. The 60/40 PLA/PBAT blend has a higher composition of PLA than PBAT and more ester bonds are available, favoring microorganism attacks, as observed in the higher weight loss of the blend compared to the neat polymers. On the other hand, weight loss was higher for the blends with and without canola oil compared to the neat polymers after the 3 and 6 months. Significant statistical differences (*p* < 0.05) were found between the fibers. The weight loss of the fibers after 3 and 6 months of composting is shown in Table 2.

According to ASTM 6400 [20] for an aerobically composted plastic, 90% of film degradation must be achieved in 180 days (6 months). According to the results, the PLA/PBAT/PPF/CO/AA with and without temperature reached this percentage (92 ± 7 and 87 ± 11%, respectively). In a previous study with PLA/PBAT nets with and without cinnamon essential oil (CEO), weight loss of 15, 7, 3, and 5% was observed for PLA, PBAT, PLA/PBAT 60/40, and PLA/PBAT/CEO, respectively, with 6.1% of CEO after 3 months of composting. At the end of the composting time (6 months) these percentages were 15, 5, 4, and 6% for PLA, PBAT, PLA/PBAT 60/40, and PLA/PBAT/CEO, respectively. After 3 months, the values were similar for neat polymers (PLA and PBAT) [21]. However, the weight loss values for PLA/PBAT were higher in this work. Among the factors that affect composting are aeration, moisture, nutrients, temperature, lignin content, polyphenols, and pH [22]. These factors could influence the kinetics of aerobic composting [23].

Ruggero et al. [24] reported a weight loss of 8% and 3% for pure PBAT and PLA films, respectively, under soil and composting conditions for 60 days. The PBAT value was similar to that found in this work after 3 months. However, the authors mentioned that the degradation of PLA increased depending on the thermophilic and mesophilic conditions, and on film thickness.

Moreover, depending on the biopolymer, lower or faster degradation can be found in soil. For example, starch degradation is faster than PLA because PLA maintains its weight for a period of 3 weeks [25]. On the other hand, a higher biodegradation rate between 80–100% has been found for cellulose-based biopolymers after 3 months [26,27], similar to the values obtained in this work for the PPF that contained the fibers. The size and structure of the lignocellulosic filler, such as PPF, influenced the degradation process. It has been reported that for pure cellulose, more accessibility points and microorganism attacks were created due to moisture and adhesion to the polymer [28].

### 3.2. Optical and SEM Observations

According to the results, the initial fibers present a smooth surface. However, after 3 months of degradation in soil, the PLA surface became rough. Dark spots could be due to microorganism colonies or residual soil/compost. As composting time increased, a color change to yellowish-brown was observed in the fibers. After 6 months, cracks and a broken fiber surface were observed as a result of biodegradation [29]. Higher fiber disintegration was observed in the PLA/PBAT/CO fiber after 3 months of decomposition. As the degradation progressed, a loss of transparency of the fibers was observed due to possible loss of crystallinity associated with PLA component hydrolysis and sample biodeterioration, as reported by Brdlík et al. [30] and Carbonell-Verdu et al. [15]. PLA/PBAT/PPF/CO/AA fibers with and without temperature showed similar degradation behavior. In Table 3, the optical surface micrographs of the fibers before and after 3 and 6 months of degradation are shown.

The SEM results (Table 4 and Table 5) revealed differences between the initial and degraded fibers. Before degradation, the neat PLA and PBAT fibers showed a smooth and homogeneous surface. As for the blends, a rougher surface can be seen. The appearance of pores or cavities is observed after degradation in all of the fibers due to physicochemical degradation and biological attack by microorganisms [10]. In PLA it has been found that during chain scission in degradation, lactic acid oligomers are formed, thereby increasing the concentration and catalytic action of carboxylic acid end groups [31]. For PBAT, chain scission of ester groups and the reaction of carboxylic groups with water in the surrounding medium occurs [32].

A porous structure with nanometric holes can be seen more clearly for PLA and PLA/PBAT blends after 3 and 6 months of degradation in Table 5. This was also observed by Ruggero et al. [24] for PLA/PBAT/starch blends. On the other hand, a “lunar surface” morphology is observed in the PLA/PBAT/CO fibers after 3 months. This can be related to the viscosity of the oil and the miscibility between the CO and the PLA/PBAT matrix [30].

In general, on the blends, pores and surface erosion are observed during the composting process due to the environmental aging of the fibers. Fiber delamination can be seen due to the progressive erosion of the material, especially in the PLA/PBAT/CO after 6 months of degradation.

In the PLA/PBAT/PPF/CO/AA fibers with and without temperature, the PPF particles are not seen in the micrographs, which is indicative of good adhesion between the PPF and the polymer matrix. Spiridon et al. [8] found a similar observation for lignin particles in a PLA matrix. Moreover, the delamination of the fiber due to erosion is not observed compared to PLA/PBAT/CO. This could be due to the AA enhancing the phase compatibility between polymers.

Comparing the PLA/PBAT/PPF/CO/AA fibers with and without temperature, higher degradation can be seen in PLA/PBAT/PPF/CO/AA. The temperature in emulsions promotes the kinetic energy and mobility of the particles, thereby increasing the collisions between them with a subsequent improvement in emulsion stability [33,34].

The detachment of fragments from the fibers and the formation of pores or cavities contributes to higher weight loss, being more evident after 3 months for the neat polymers. In Table 5, more cavities can be seen in PLA compared to PBAT, with a weight loss of 19 and 7%, respectively (see Table 2). The same behavior was found for PLA fibers under composting by Feijoo et al. [28].

Table 4 and Table 5 show the SEM micrographs of the surface morphology of the fibers before and after 3 and 6 months of degradation for 100 and 5 k magnification, respectively.

### 3.3. Fourier Transform Infrared Spectroscopy–Attenuated Total Reflection (FTIR-ATR)

In Figure 1, the FTIR-ATR spectra of the fibers show the changes that occurred on their surface due to the degradation process.

At the initial time, the main characteristic absorption peaks for PLA, PBAT, and PLA/PBAT have already been reported [13,24,30,35,36]. The main PLA bands and peaks (see Figure 1a) are between 3760 and 3115 cm^−1^, assigned to the -OH groups, and between 3000 and 2775 cm^−1^, corresponding to -CH deformations (symmetric and asymmetric bends). The sharp peak at 1740 cm^−1^ is attributed to C=O from the ester groups; at 1450 cm^−1^ to -CH in CH_3_; at 1272 cm^−1^ it is related to C-O stretching; at 1180 for C-O stretching in the -CH-O group; and at 1079 cm^−1^ it is related to C-O-C stretching. For PBAT, the main peaks observed in Figure 1b are similar to those observed for PLA, due to the similar chemical structure except for the band between 1210 and 1000 cm^−1^, which is related to the benzene rings, and the peak at 725 cm^−1^, for the bending vibration of CH-plane in the benzene ring [37]. No differences were observed on the bands for the blends after PBAT incorporation into PLA and the addition of CO, PPF, and the influence of temperature.

After biodegradation, the following changes were observed. In PBAT and the polymer blends, an increase in the band between 3760 and 3115 cm^−1^ corresponding to the -OH groups with degradation time was observed (Figure 1b–f). This is related to bond scission and -OH formation, characteristic of the hydrolysis process that occurs in the fibers during composting [10,30]. 

A detail of the FTIR spectra of the carboxyl groups (-C=O) in the region between 1800 and 1600 cm^−1^ to trace the degradation process is shown in Figure 2.

A decrease in the peak at 1740 cm^−1^ related to the stretching vibrations of -C=O can be seen as a result of chain scission during degradation [36,38] as well as in the band at around 1100 cm^−1^, correlated to -C-O groups as shown in Figure 3 [36].

Moreover, the absorption band at 1269 cm^−1^ showed a decrease for the PLA/PBAT/PPF/CO/AA with and without temperature as observed in Figure 4a,b. This band is related to hemicelluloses biodegradation from acetyl and carboxylic vibration of the carbohydrate component of plant cell walls. It has been reported that lignin absorbs at a wavelength near 1269 cm^−1^ [39]. The FTIR-ATR spectrum of PPF is shown in Figure 4c. The decrease in this peak was only observed in the PLA/PBAT/PPF/CO/AA with and without temperature fibers. The decrease in this absorption band could be related to the higher weight loss observed in Table 2.

### 3.4. Differential Scanning Calorimetry (DSC)

The results of thermal analysis by DSC for the fibers before and after degradation are shown in Figure 5 and Figure 6. The cooling and heating thermograms are shown in these two figures, also.

In Figure 5a, the glass transition temperature (T_g_) of PLA can be seen at around 61–62 °C. For PBAT in Figure 5b, the crystallization temperature (T_c_) increased for the fibers as the degradation time progressed with values of 95.6 °C (initial), 97.4 °C (3 months), and 98.1 °C (6 months). For the blends, the same behavior was observed. As the degradation time increased, the T_c_ increased as well as the PLA T_g_ (Figure 5c,d). On the other hand, when PPF and AA were added to the blends, the T_c_ value shifted to higher values, being highest for the PLA/PBAT/PPF/CO/AA _T fiber. For example, after 6 months of degradation the T_c_ value of the PLA/PBAT/CO fiber was 122.4 °C and of the PLA/PBAT/PPF/CO/AA _T fiber it was 125.8 °C with a difference of almost 3 °C. For the PLA/PBAT/PPF/CO/AA fiber, a similar value to that of PLA/PBAT/CO was found (122.5 °C). The increase in T_c_ is associated with interaction between phases in the polymer blends. For the PLA/PBAT/PPF/CO/AA and PLA/PBAT/PPF/CO/AA fibers with and without temperature, the stiffening effect of the PPF and the enhancement of blend compatibility due to AA influenced the crystallization process of the blend. The increase in T_c_ is indicative of an increase in crystal size [26].

From Figure 6, the second heating of the fibers before and after degradation can be observed.

The DSC curves corresponding to the second heating can be seen in Figure 6. In Figure 6a, the T_g_ and T_m_ of neat PLA were observed at 60.5 and 148.4 °C, respectively. Some authors have reported T_g_ values between 59.1 °C and 62.4 °C and T_m_ values between 147.8 °C and 154 °C [13,16,37]. While the T_m_ of PLA before and after degradation remained constant (60.4–60.6 °C), the melting showed some differences. In the detail of Figure 7a, before degradation, only one peak at 148.4 °C can be seen. After degradation the curve splits into two peaks showing T_m_ at 148.5 °C and 153.5 °C after 3 months and 148.4 °C and 154.3 °C after 6 months. While the main melting peak (148.4–148.5 °C) remained steady, the area and T_m_ of the new one increased progressively. This higher temperature component is attributed to new crystalline phases formed by smaller chains resulting from the degradation process [31,37].

The T_g_ value for PBAT was found at −33.1 °C (Figure 6b), close to the −34 °C reported by Turek et al. [40]. On the other hand, the T_m_ (129.8 °C) was slightly above those reported between 120 °C and 124.1 °C [13,16,34]. As for PLA, the T_m_ of PBAT increased with degradation time (129.8 °C neat, 130.5 °C after 3 months, and 131.2 °C after 6 months).

The thermal behavior of the PLA component in the PLA/PBAT blend (Figure 6c) is slightly different when compared to that of neat PLA, as shown in Figure 7a,b. Both T_g_ and T_m_ decreased by 4 and 1.9 °C, respectively, and a higher melting temperature (153.2 °C) developed. These findings suggest the interaction between PLA and PBAT in the blend. Additionally, both the T_g_ and T_m_ values increased with degradation time: 56.5, 58.7 and 59.1 °C and 146.5, 147.5 and 147.5 °C initially and after 3 and 6 months, respectively, and approached those of neat PLA. This result can be interpreted as the selective degradation of either or both PBAT or PBAT-altered PLA regions.

For the PLA/PBAT/CO, PLA/PBAT/PPF/CO/AA, and PLA/PBAT/PPF/CO/AA_T fibers, a similar behavior was observed, as shown in Figure 6d–f. First, there was a reduction in T_g_ for both the PBAT and PLA phases with respect to the PLA/PBAT blend, which is attributable to the plasticizing effect of canola oil. Second, there was a more noticeable increment in T_g_ values with degradation time, which points to the leaching of the additives along the degradation.

Between the T_g_ and T_m_, a cold crystallization (endothermic peak) is observed for PLA and the blends. The cold crystallization peak for the PLA (Figure 6a) was at around 115 °C and for the PLA/PBAT blend (Figure 6c) at around 113 °C. For the other blends, the cold crystallization temperature (T_cc_) was lower compared to PLA and, as degradation time increased, the T_cc_ increased. Values between 117 °C and 120 °C have been reported in the literature [15,30]. Nomadolo et al. [41] and Vasile et al. [38] found higher values of T_cc_ for PLA compared to the blends, as shown in this work. This is correlated with the presence of a nucleating agent in the blend composition accelerating the crystallization and decreasing the T_cc_ [37,41].

Comparing the fibers as the degradation process progresses, it has been reported that the crystalline domains increased during degradation. These domains act as networks or cross-linking points, limiting the mobility of PLA. Only amorphous chains with high molecular mobility undergo rearrangement and crystallize at low temperature resulting in cold crystallization. On the other hand, the chains with low mobility also crystallize, but at higher temperatures [3,38,42].

The detail of Figure 8a shows that for the PLA/PBAT/PPF/CO/AA blend, a similar T_cc_ value was obtained for the initial fiber and after 3 months of degradation (105.0 °C and 104.3 °C, respectively). After 6 months of degradation, this value increased by 8.5 °C (112.6 °C). For the PLA/PBAT/PPF/CO/AA_T fiber, there was a displacement of T_cc_ to higher values from 108.5 °C (initial) to 110.5 °C (3 months) and 113.2 °C (6 months). As seen in Figure 8b, after degradation, a similar T_cc_ value was reached after both 3 and 6 months. This could be related to the adhesion or uniform dispersion of the PPF with the use of temperature in the emulsion limiting the blend’s segmental motion [43,44].

DSC showed that the crystallization temperature (T_c_) increased for the fibers as the degradation time progressed, as well as when PPF and AA were incorporated into the blend. For the melting, new crystalline phases resulting from the degradation process were found. Regarding cold crystallization, the most important change was observed for the PLA/PBAT/PPF/CO/AA fiber with an increase in T_cc_ after 6 months of degradation.

### 3.5. Mechanical Properties

The mechanical properties of the fibers are reported in Table 6 and also illustrated in Figure 9. For the initial fibers, PLA showed a higher Young´s modulus and tensile strength but lower elongation at break than PBAT, as shown in Table 6. A decrease in the tensile strength of PLA fibers in compost over time has been reported by Borelbach et al. [10]. In the case of the PLA/PBAT blend, the modulus and tensile strength were close to averages of both components. However, the deformation was similar to that of neat PLA; therefore, the incorporation of PBAT into PLA did not increase the flexibility of PLA. By adding CO, Young´s modulus and tensile strength decreased compared to PLA/PBAT. On the other hand, the elongation at break was improved. It has been reported in the literature that CO improves the elongation at break [45]. On the other hand, when AA (PLA/PBAT/PPF/CO/AA) and temperature (PLA/PBAT/PPF/CO/AA_T) were used, an increase in Young´s modulus, tensile strength, and a decrease in elongation at break was observed compared to PLA/PBAT/CO. This can be attributed to the stiffness of the PPF and the improvement of the interfacial adhesion and stress transfer between the polymer matrix and the PPF [44]; therefore, ductility and toughness were enhanced with the addition of PPF [43]. However, after 3 and 6 months of degradation, those properties decreased, indicating a restriction in segmental mobility [8] and an increase in crystalline regions with degradation time [38], as clearly seen in Figure 9. This was also observed via the increase in T_cc_ and T_m_ in the DSC thermograms (Figure 6).

After 3 and 6 months of degradation, the trend was the same for the initial neat polymers and the PLA/PBAT blend. However, for the PLA/PBAT/CO fibers, Young´s modulus was higher compared to the PLA/PBAT fibers, the opposite of the initial ones. The tensile strength and elongation at break values were similar. After AA incorporation (PLA/PBAT/PPF/CO/AA), Young´s modulus decreased as well as the tensile strength and elongation at break. Finally, for the PLA/PBAT/PPF/CO/AA_T fibers, the modulus decreased and the tensile strength and elongation at break remained almost the same compared to the PLA/PBAT/PPF/CO/AA fibers.

As the degradation time increased, there was a decrease in the mechanical properties of the PLA, PBAT, and PLA/PBAT/PPF/CO/AA_T fibers. This difference was more noteworthy when comparing the initial and degraded fibers, being similar for the 3- and 6-month degraded ones. This decrease in mechanical properties can be seen in SEM micrographs (Table 5), in which the fiber delamination, holes, and cracks could be observed in the blend fibers, associated with the weight loss (Table 2). It has been reported by several authors [46,47,48] that the mechanical properties decreased after degradation in PLA, PBAT, and their blends.

The incorporation of PPF and AA into the blends modifies interactions between the components. Moreover, when temperature was used for the emulsion preparation, lower weight loss was observed compared to the PLA/PBAT/PPF/CO/AA fibers, so perhaps the PPF loss was hindered by these interactions [38]. As can be observed from the results, degradation is a complex process that depends on the environmental conditions [49,50,51].

Statistically, the main significant differences (*p* ≤ 0.05) between compositions and degradation time were obtained for the mechanical properties of PLA and PBAT.

## 4. Conclusions

This study investigated the effect of adding PPF to a biodegradable PLA/PBAT polymer blend. After 6 months of composting, the studied fibers containing PF were degraded by 87 ± 11%. Morphological changes were observed by optical microscopy and SEM. The analysis of the FTIR-ATR spectra showed a decrease in the absorption bands associated with polymer matrix degradation. The thermal stability and mechanical properties decreased as degradation progressed. The PLA/PBAT/PPF/CO/AA composites resulted in accelerated biodegradation, useful for the disposal of any material.

## Figures and Tables

**Figure 1 polymers-15-04164-f001:**
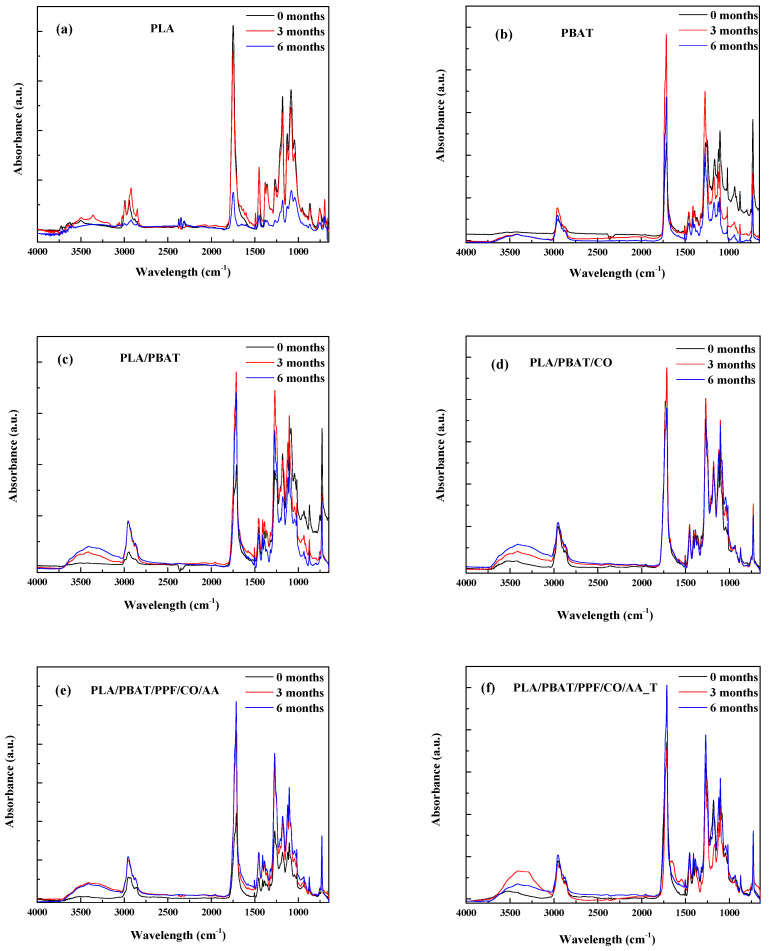
FTIR spectra of (**a**) PLA, (**b**) PBAT, (**c**) PLA/PBAT, (**d**) PLA/PBAT/CO, (**e**) PLA/PBAT/PPF/CO/AA, (**f**) PLA/PBAT/PPF/CO/AA_T.

**Figure 2 polymers-15-04164-f002:**
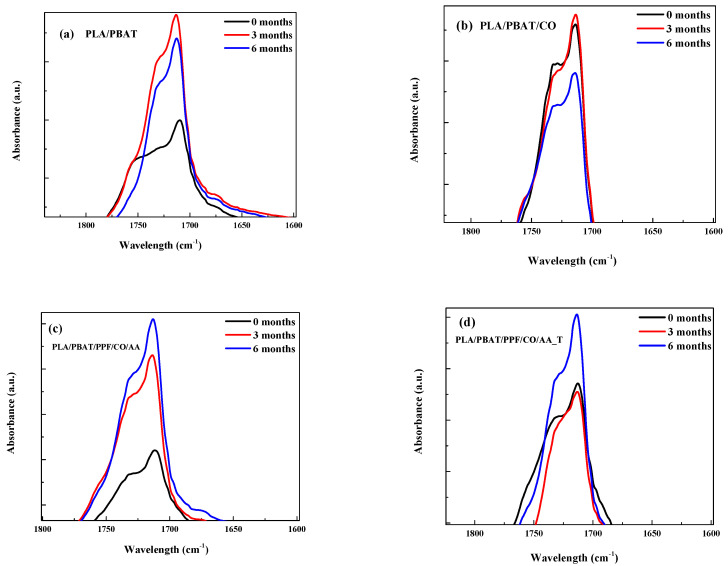
Detailed FTIR spectra of (**a**) PLA/PBAT, (**b**) PLA/PBAT/CO, (**c**) PLA/PBAT/PPF/CO/AA, (**d**) PLA/PBAT/PPF/CO/AA_T at 1740 cm^−1^.

**Figure 3 polymers-15-04164-f003:**
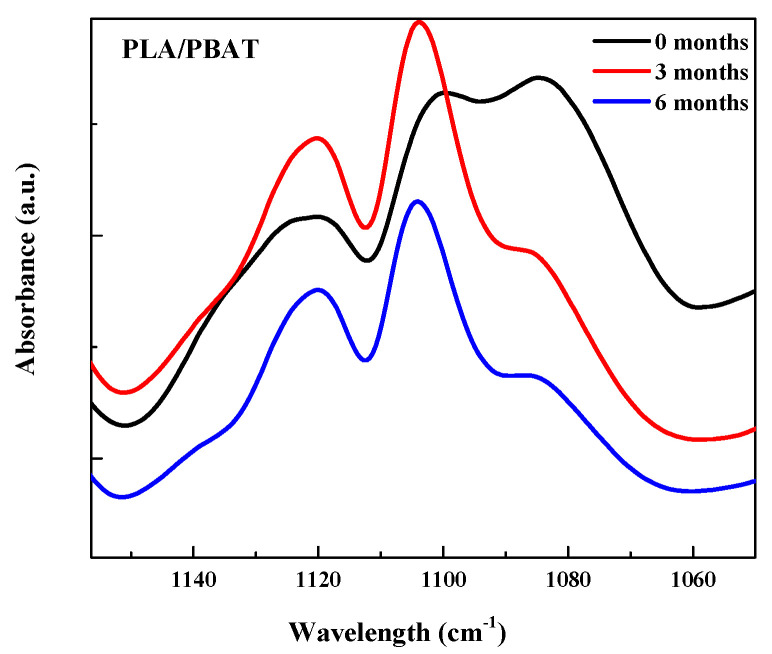
Detailed FTIR spectra of PLA/PBAT at 1110 cm^−1^.

**Figure 4 polymers-15-04164-f004:**
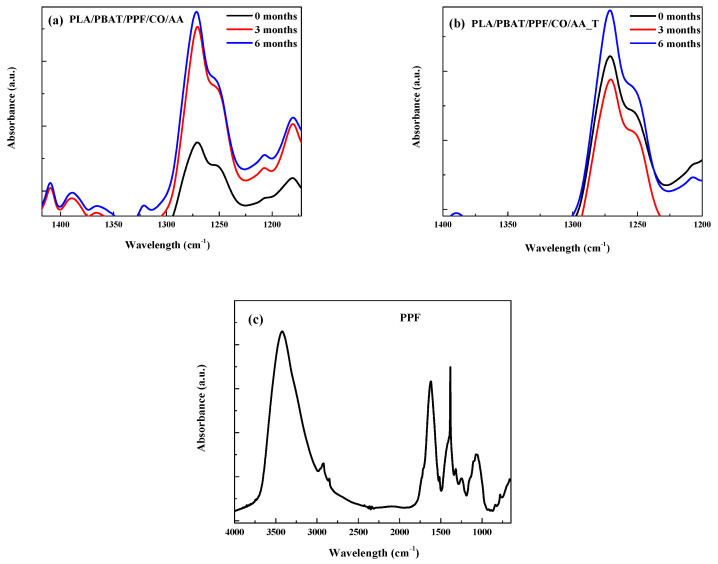
Detailed FTIR spectra of (**a**) PLA/PBAT/PPF/CO/AA, (**b**) PLA/PBAT/PPF/CO/AA_T, and (**c**) PPF at 1269 cm^−1^.

**Figure 5 polymers-15-04164-f005:**
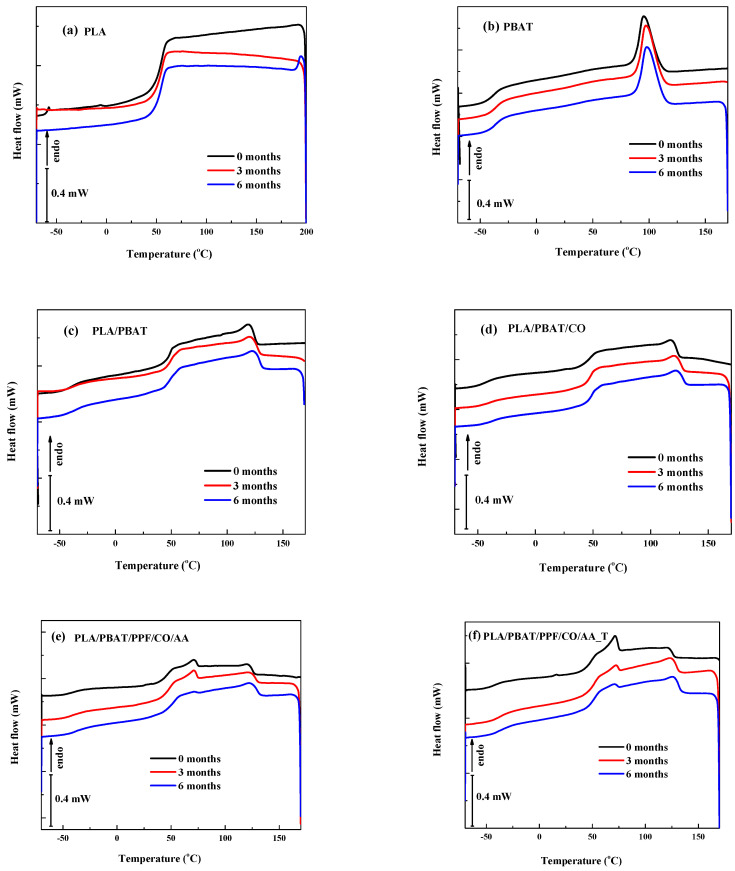
Cooling DSC thermogram of neat polymers and blend fibers before and after degradation.

**Figure 6 polymers-15-04164-f006:**
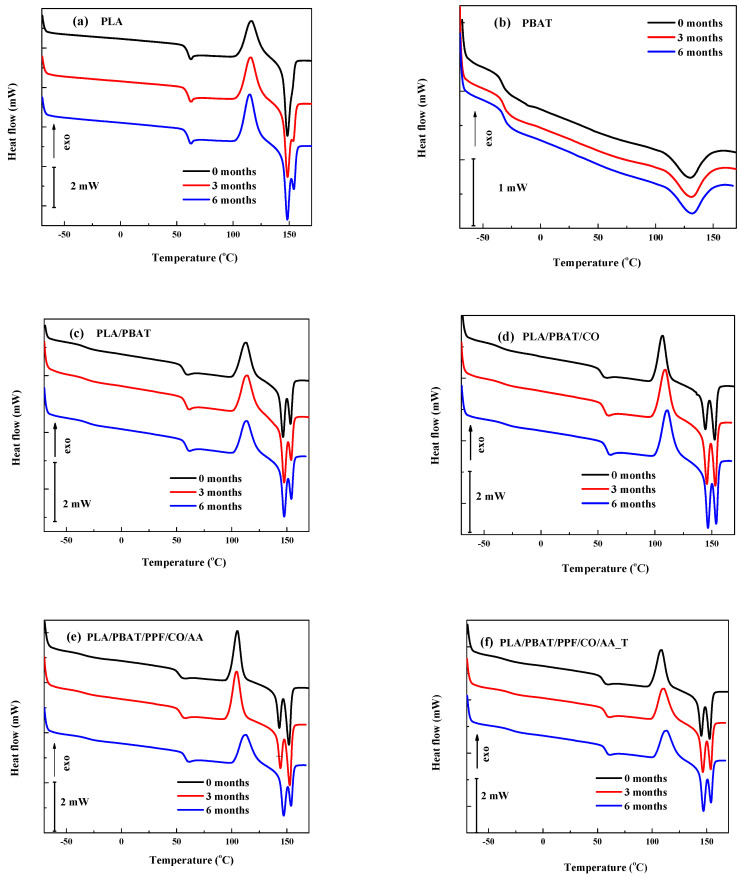
Second heating DSC thermogram of neat polymers and blend fibers before and after degradation.

**Figure 7 polymers-15-04164-f007:**
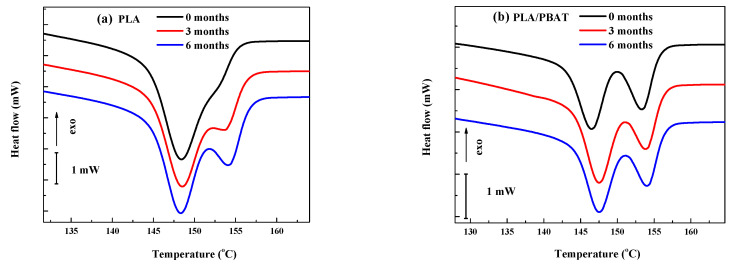
Details of the second heating DSC thermogram for (**a**) PLA and (**b**) the PLA/PBAT blend before and after degradation.

**Figure 8 polymers-15-04164-f008:**
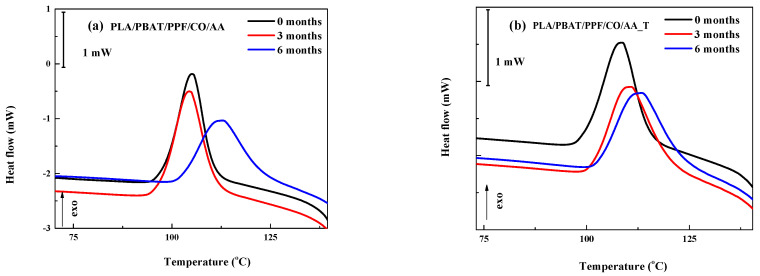
Details of the second heating DSC thermogram for (**a**) PLA/PBAT/PPF/CO/AA and (**b**) PLA/PBAT/PPF/CO/AA_T fibers before and after degradation.

**Figure 9 polymers-15-04164-f009:**
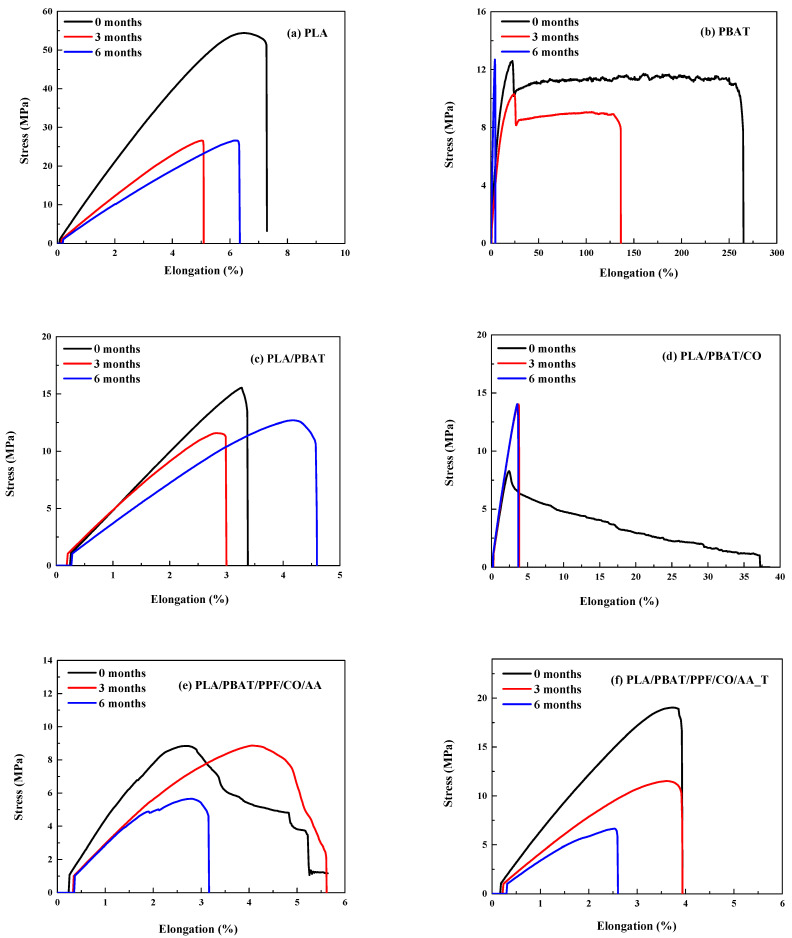
Stress–strain curves for (**a**) PLA, (**b**) PBAT, (**c**) PLA/PBAT, (**d**) PLA/PBAT/CO, (**e**) PLA/PBAT/PPF/CO/AA, and (**f**) PLA/PBAT/PPF/CO/AA_T before and after degradation.

**Table 1 polymers-15-04164-t001:** Composition of the biodegradable fibers.

Fiber	Polymer (%)	CO (%)	PPF (%)	AA (%)
PLA	100	0	0	0
PBAT	100	0	0	0
PLA/PBAT	100	0	0	0
PLA/PBAT/CO	96	4	0	0
PLA/PBAT/PPF/CO/AA	86	4	3	7
PLA/PBAT/PPF/CO/AA_T	86	4	3	7

**Table 2 polymers-15-04164-t002:** Weight loss (%) of the fibers after 3 and 6 months of composting in soil.

Weight Loss Percentage (%) *		
Fiber	3 Months	6 Months
PLA	20 ± 1 ^B^	45 ± 13 ^A^
PBAT	8 ± 1 ^A^	49 ± 12 ^A^
PLA/PBAT	44 ± 7 ^C^	65 ± 11 ^B^
PLA/PBAT/CO	25 ± 1 ^B^	64 ± 3 ^A^
PLA/PBAT/PPF/CO/AA	62 ± 4 ^D^	92 ± 7 ^C^
PLA/PBAT/PPF/CO/AA_T	56 ± 9 ^D^	87 ± 11 ^BC^

* Different letters represent statistical differences between the fibers by Tukey´s test (*p* < 0.05).

**Table 3 polymers-15-04164-t003:** Optical surface micrographs of the fibers initially and after 3 and 6 months of composting in soil.

Fiber	Initial	3 Months	6 Months
PLA	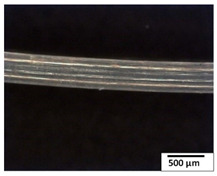	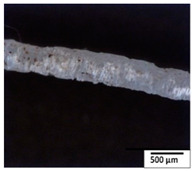	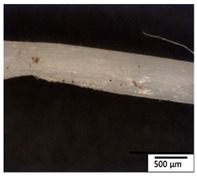
PBAT	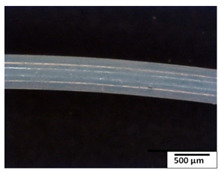	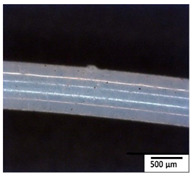	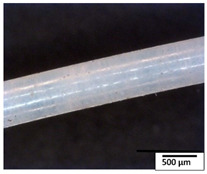
PLA/PBAT	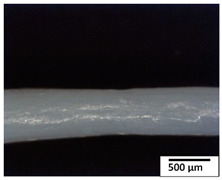	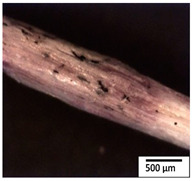	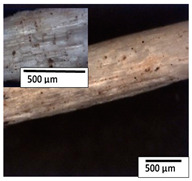
PLA/PBAT/CO	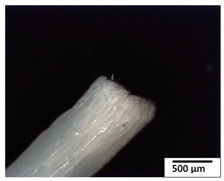	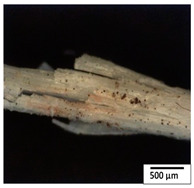	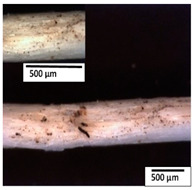
PLA/PBAT/PPF/CO/AA	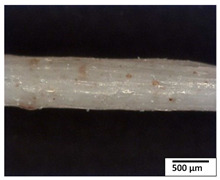	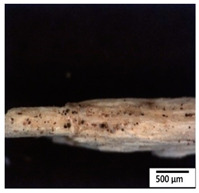	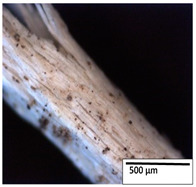
PLA/PBAT/PPF/CO/AA_T	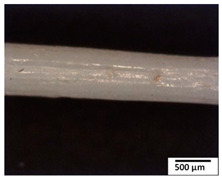	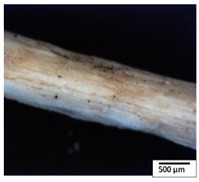	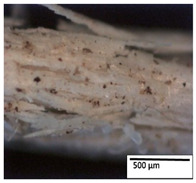

**Table 4 polymers-15-04164-t004:** SEM micrographs of the fiber’s surface morphology initially and after 3 and 6 months of composting in soil (magnification ×100).

Fiber	Initial	3 Months	6 Months
PLA	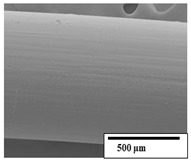	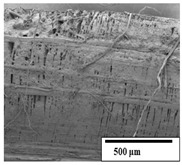	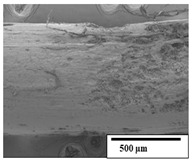
PBAT	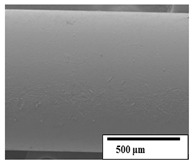	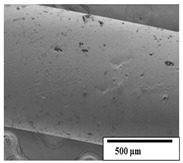	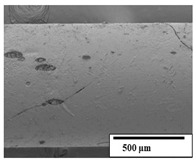
PLA/PBAT	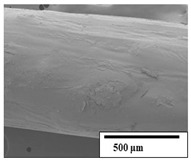	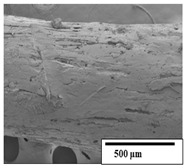	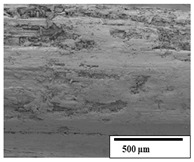
PLA/PBAT/CO	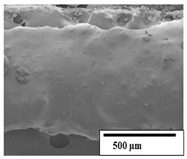	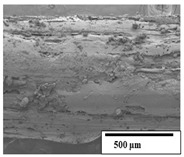	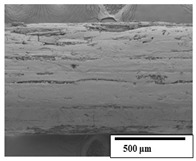
PLA/PBAT/PPF/CO/AA	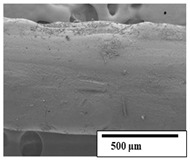	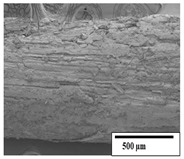	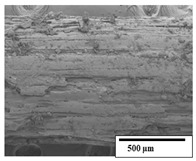
PLA/PBAT/PPF/CO/AA_T	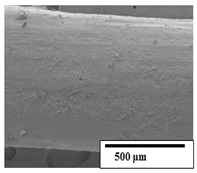	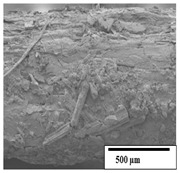	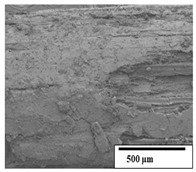

**Table 5 polymers-15-04164-t005:** SEM micrographs of the fiber’s surface morphology initially and after 3 and 6 months of composting in soil (magnification ×5k).

Fiber	Initial	3 Months	6 Months
PLA	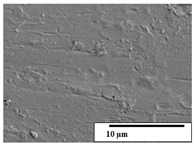	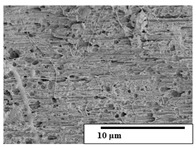	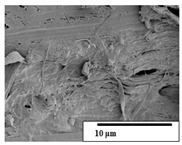
PBAT	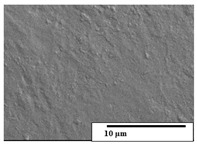	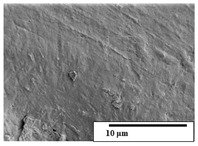	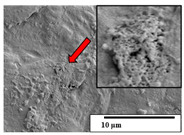
PLA/PBAT	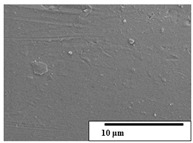	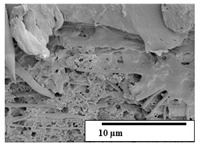	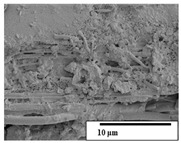
PLA/PBAT/CO	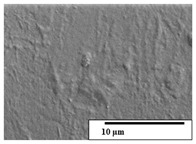	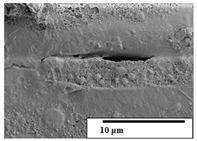	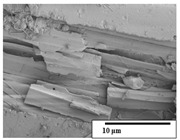
PLA/PBAT/PPF/CO/AA	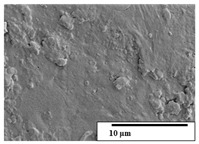	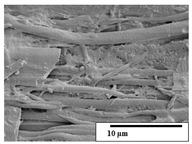	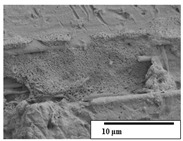
PLA/PBAT/PPF/CO/AA_T	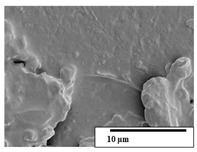	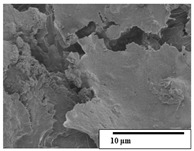	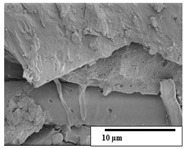

**Table 6 polymers-15-04164-t006:** Mechanical properties of PLA, PBAT, PLA/PBAT, PLA/PBAT/CO, PLA/PBAT/PPF/CO/AA, and PLA/PBAT/PPF/CO/AA_T before and after degradation.

Mechanical Properties			
Fiber Composition and Time of Composting	Young’s Modulus (MPa) *	Tensile Strength (MPa) *	Elongation at Break (%) *
Initial			
PLA	1109 ± 47 ^Cg^	55 ± 2 ^Ce^	6 ± 1 ^Aa^
PBAT	115 ± 10 ^Aa^	11 ± 1 ^Abc^	224 ± 31 ^Bc^
PLA/PBAT	507 ± 89 ^Bcdef^	21 ± 4 ^Bd^	5 ± 1 ^Aa^
PLA/PBAT/CO	412 ± 131 ^Bbcd^	8 ± 2 ^Ab^	40.87 ± 5 ^ABc^
PLA/PBAT/PPF/CO/AA	435 ± 83 ^Bbcde^	9 ± 1 ^Ab^	3 ± 1 ^Aa^
PLA/PBAT/PPF/CO/AA_T	552 ± 89 ^Bdef^	18 ± 3 ^Bcd^	5 ± 1 ^Aa^
3 months			
PLA	605 ± 61 ^Cf^	25 ± 2 ^Cde^	5 ± 1 ^Aa^
PBAT	98 ± 5 ^Aa^	11 ± 1 ^Abc^	130 ± 34 ^Bb^
PLA/PBAT	430 ± 52 ^Bbcde^	16 ± 4 ^Bcd^	5 ± 1 ^Aa^
PLA/PBAT/CO	497 ± 88 ^Bcdef^	14 ± 3 ^ABc^	3 ± 1 ^Aa^
PLA/PBAT/PPF/CO/AA	463 ± 87 ^Bbcdef^	11 ± 3 ^ABbc^	3 ± 1 ^Aa^
PLA/PBAT/PPF/CO/AA_T	385 ± 55 ^Bbc^	11 ± 3 ^Abc^	3 ± 1 ^Aa^
6 months			
PLA	562 ± 71 ^Def^	27 ± 6 ^Cde^	6 ± 1 ^Aa^
PBAT	98 ± 5 ^Aa^	11 ± 1 ^ABbc^	5 ± 1 ^Aa^
PLA/PBAT	397 ± 31 ^BCbc^	13 ± 3 ^ABbc^	4 ± 1 ^Aa^
PLA/PBAT/CO	455 ± 117 ^Cbcde^	13 ± 5 ^Bbc^	4 ± 1 ^Aa^
PLA/PBAT/PPF/CO/AA	413 ± 39 ^BCbcd^	7 ± 1 ^Aa^	3 ± 1 ^Aa^
PLA/PBAT/PPF/CO/AA_T	340 ± 29 ^Bb^	7 ± 2 ^ABb^	3 ± 1 ^Aa^

* Different letters represent statistical differences between compositions (upper case letters) and months (lower case letters) by Tukey test (*p* < 0.05).

## Data Availability

Data available on request from the authors.

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
