# Peer review of "Biodegradability Assessment of Prickly Pear Waste–Polymer Fibers under Soil Composting"

_polymers, 2023, doi:10.3390/polym15204164_

Round 1

Reviewer 1 Report

I think this is well-written manuscript with conclusions well-supported by the experimental results. It can be published after some minor comments below are addressed.

-please provide standard deviations for all experimental data obtained in the manuscript.

-please eliminate excessive digits after decimal in Table 2 and 6

-please proofread the manuscript to eliminate some typos

-please check Table 1. I believe composition of PLA/PBAT/CO is not correct

-please comment/elaborate on significantly higher weight loss percentage for PLA/PBAT in comparison to PLA and PBAT (or provide references) (Table 2)

I recommend the manuscript for publication in Polymers after those comments are addressed

Author Response

Dear reviewer 1, thank you for your useful comments. Changes are highlighted in yellow in the manuscript. Please find responses to your questions below.

Comments and Suggestions for Authors

QUESTION:

-please provide standard deviations for all experimental data obtained in the manuscript.

ANSWER:

Standard deviations for all experimental data were added.

QUESTION:

-please eliminate excessive digits after decimal in Table 2 and 6

ANSWER:

Excessive decimal digits were eliminated form Table 2 and 6.

QUESTION:

-please proofread the manuscript to eliminate some typos

ANSWER:

English was revised and some changes were made to the manuscript.

QUESTION:

-please check Table 1. I believe composition of PLA/PBAT/CO is not correct

ANSWER:

You were right. It was corrected.

QUESTION:

-please comment/elaborate on significantly higher weight loss percentage for PLA/PBAT in comparison to PLA and PBAT (or provide references) (Table 2)

ANSWER:

A comment was added to the manuscript.

PLA and PBAT degradation efficiency is related to the difference in polymer structure and action of the soil microorganisms [Zhang et al., 2022]. Some authors have demonstrated for PLA/PBAT blends that during composting, there is a selectivity to degrade the PLA phase with the formation of a porous 3D network [Tabasi et al., 2015] as observed in Table 5. In the literature, it has been reported that the lower weight loss or PBAT can be attributed to the presence of benzene rings [Liu et al., 2022]. During degradation, microorganisms attack the ester bond in PLA and PBAT, with the generation of small molecules or oligomers, which are subsequently metabolized by these microorganisms [Zhang et al., 2022]. The 60/40 PLA/PBAT blend has a high composition of PLA than PBAT and more ester bonds are available favoring microorganism’s attack as observed for a higher weight loss of the blend compared to the neat polymers.

Zhang, Y.; Gao, W.; Mo, A.; Jiang, J., He, D. Degradation of Polylactic Acid/Polybutylene Adipate Films in Different Ratios and the Response of Bacterial Community in Soil Environments. Environm. Pollut. 2022, 313, 120–167. https://doi.org/10.1016/j.envpol.2022.12016.

Tabasi, R. Y.; Ajji, A. Selective Degradation of Biodegradable Blends in Simulated Laboratory Composting. Polym. Degrad. Stab. 2015, 120, 435–442. https://doi.org/10.1016/j.polymdegradstab.2015.07.020.

Liu, B.; Guan, T.; W. G.; Fu, Y.; Weng, Y. Behavior of Degradable Mulch with Poly (Butylene Adipate-coTerephthalate) (PBAT) and Poly (Butylene Succinate) (PBS) in Simulation Marine Environment. Polymers (Basel) 2022, 14, 1515. https:// doi.org/10.3390/polym14081515.

Reviewer 2 Report

The study of the biodegradation process in the soil of PLA/PBAT polymer fibers added with prickly pear flour as agro-industrial waste, adipic acid as a compatibilizer, and canola oil as a plasticizer is interesting.

The work presented here is exciting but requires major revisions before it can be accepted for publication. My comments are as follows:

1. The authors need to consider using other characterization techniques, such as X-ray diffraction calorimetry for crystallinity, and surface properties of prickly pear polymer fibers (at least some best results before degradation and after degradation).

2. You have mentioned only the tensile strength, and elongation at break data in tabulated form. Please provide the figures of stress vs elongation data for better understanding of the readers (some of the best results of before and after degradation).

3. You have not mentioned how to calculate the Young's modulus (MPa) from tensile strength. Please provide the calculation procedure of Young´s modulus.

Author Response

Dear reviewer 2, thank you for your useful comments. Changes are highlighted in green in the manuscript. Please find responses to your questions below.

Comments and Suggestions for Authors

The study of the biodegradation process in the soil of PLA/PBAT polymer fibers added with prickly pear flour as agro-industrial waste, adipic acid as a compatibilizer, and canola oil as a plasticizer is interesting.

The work presented here is exciting but requires major revisions before it can be accepted for publication. My comments are as follows:

QUESTION:

  1. The authors need to consider using other characterization techniques, such as X-ray diffraction calorimetry for crystallinity, and surface properties of prickly pear polymer fibers (at least some best results before degradation and after degradation).

ANSWER:

Thanks for your useful suggestion. It would have been good to do thermal diffraction given the cold crystallization phenomena observed. To our knowledge, there is equipment able to perform in situ Differential Scanning Calorimetry measurements in an X-Ray Diffractometer. In our case the DSC was done separately. We would apologize for not having done thermodiffraction (in future work)

The calculation of % crystallinity (Xc) is complex. The melting peak of PBAT falls in the middle of cold crystallization and melting of PLA. It is a very broad peak (taking as a reference what was seen in the PBAT sample) included in the enthalpy calculation obtained from the DSC. On the other hand, it is not common to see that PLA is practically amorphous under the preparation conditions (Xc=1.3%) and then, for the blends, the enthalpy values obtained gave higher crystallinity. As an approximation, from DSC, the theoretical fusion enthalpy of a 100% crystalline blend containing 60% PLA and 40% PBAT was calculated (101.4 J/g). Both, PLA and PBAT and their blends have been characterized in the literature so that the PLA melting peaks can be assigned. This would give an average crystallinity without specifying whether it is due to PLA or PBAT. Therefore, it would be necessary to do modulated DSC to separate reversible events (fusion) from irreversible events (cold crystallization). The crystallinity values calculated in this way (average Xc) are detailed below in Table 1. If it were assigned to the PLA phase, as is commonly do in the literature, the values would be those in Table 2 (Xc PLA).  The percentage of crystallinity was calculated from the second heating using Equation: Xc (%) = 100 × âˆ†Hm / (f × âˆ†H0m) where ∆Hm is the enthalpy of fusion (J/g), f is the weight fraction of PLA or PBAT, and ∆Hm 0 is the theoretical enthalpy of fusion for a 100% crystalline polymer, 93 J/g for PLA and 114 J/g for PBAT [Al-Itry et al., 2012].

Al-Itry, R.; Lamnawar, K.; Maazouz, A. Improvement of thermal stability, rheological and mechanical 710 properties of PLA, PBAT and their blends by reactive extrusion with functionalized epoxy. Polym. Degrad. Stabil. 2012, 97, 1898–1914. https://doi.org/10.1016/j.polymdegradstab.2012.06.028.

Average Xc (%)

Initial

3 months

6 months

PLA

1.3

1.1

1.3

PBAT

9.9

10.2

10.2

PLA/PBAT

5.3

5.2

5.4

PLA/PBAT/CO

4.5

5.0

5.5

PLA/PBAT/PPF/CO/AA

4.3

5.9

5.3

PLA/PBAT/PPF/CO/AA_T

5.1

5.9

5.8

No significant change was observed for the average Xc (%) with the degradation time for neat polymers and the PLA/PBAT sample. However, when the CO, PPF and AA are incorporated to fiber composition, an increase in crystallinity as degradation progresses was observed.

PLA Xc (%)

Initial

3 months

6 months

PLA

1.3

1.1

1.3

PBAT

9.9

10.2

10.2

PLA/PBAT

9.6

9.4

9.9

PLA/PBAT/CO

8.1

9.1

9.9

PLA/PBAT/PPF/CO/AA

7.9

10.7

9.6

PLA/PBAT/PPF/CO/AA_T

9.4

10.7

10.5

For the PLA Xc (%) similar behavior was observed compared to that of average Xc (%) for neat polymers and the PLA/PBAT sample. Moreover, for the PLA/PBAT/CO, PLA/PBAT/PPF/CO/AA and the PLA/PBAT/PPF/CO/AA_T fibers, an increase in crystallinity as degradation progresses was observed.

An increase in crystallinity as composting progresses is an indicative of faster degradation of the amorphous regions by soil microorganisms (Fu et al., 2020). This agrees with the weight loss observed for the fibers (Table 2). Due to the uncertainty regarding separation of reversible events (fusion) from irreversible events (cold crystallization), this part was not included in the manuscript.

Regarding surface properties, SEM and FTIR are important characterization techniques to observe changes during degradation. Maybe hydrophobicity could be very useful; however, it would be difficult to measure in such a fine fiber, since the drop of water is tiny.

QUESTION:

  1. You have mentioned only the tensile strength, and elongation at break data in tabulated form. Please provide the figures of stress vs elongation data for better understanding of the readers (some of the best results of before and after degradation).

ANSWER:

Figures of stress-strain for the fibers were included in the manuscript. This suggestion was very useful, in the graph it can be seen more clearly the effect of the degradation process in the decreased of the mechanical properties. Moreover, two numerical mistakes were corrected in Table 6.

QUESTION:

  1. You have not mentioned how to calculate the Young's modulus (MPa) from tensile strength. Please provide the calculation procedure of Young´s modulus.

ANSWER:

The Young´s modulus was obtained from the slope of the linear region on the low deformation side of the stress-strain curve.

Round 2

Reviewer 2 Report

Recommendation: Publish as is; no revisions needed.

Comments:

After carefully reading the revised manuscript and point-by-point response to reviewers' comments, I can fully understand the authors' argument and purpose. Thus, I recommend this paper for publication without further modification.